# Z-Cache: Accelerating Diffusion Transformers via Self-Reflection

## Abstract

Diffusion transformers have become the most powerful models for visual generation, but still suffer from massive computation costs. To solve this problem, feature caching has been proposed to cache the features of diffusion models in the previous computation steps and then reuse them in the following caching steps, which brings significant acceleration but also degradation in generation quality. To address this problem, this paper proposes Z-cache as a feature caching method that can maintain high-quality generation through self-reflection. Concretely, we observe that *the error from feature caching tends to be sharply reduced after each full computation.* Based on this observation, Z-Cache is designed to first predict the features in the future caching steps and then perform a full computation. After that, Z-Cache returns to the caching steps and re-predicts them based on the previous and the current computation steps, which brings correction in features. Experiments demonstrate that with *Z-Cache*, diffusion transformers achieve comparable generation quality to the original model but with faster inference speed, for instance, **5.53×** acceleration on FLUX-dev for text-to-image generation. *Our codes have been released in the supplementary materials and will be released on GitHub.*

## 1 Introduction

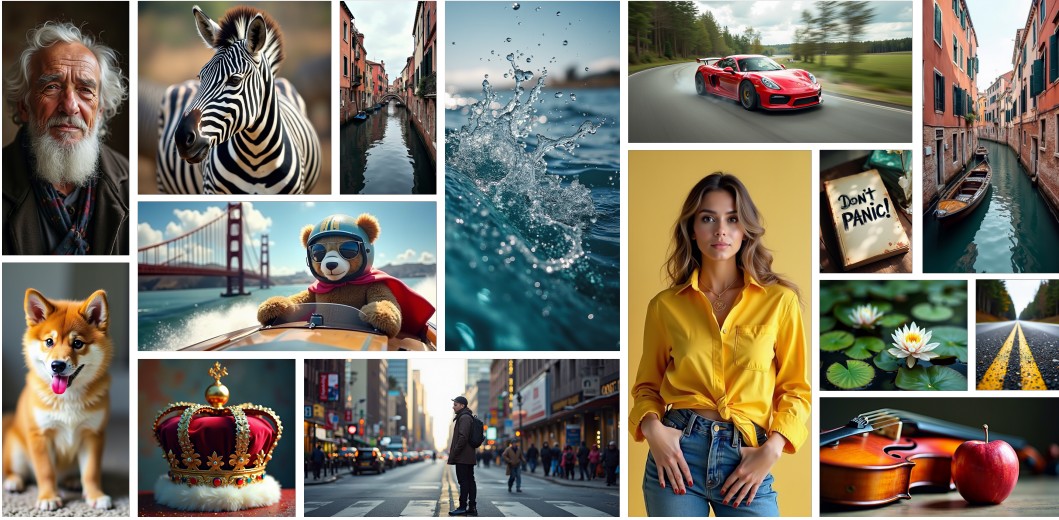

Figure 1: Images sampled by 5.53× accelerated FLUX.1-dev with *Z-Cache*.

Researchers have developed numerous methodologies for generating high-quality data across diverse modalities. Diffusion models (Ho et al., 2020; Song et al., 2021; Liu et al., 2023) have garnered significant attention due to their remarkable capacity for synthesizing intricate details, surpassing many existing approaches in fidelity and realism. However, despite their advantages, diffusion models face a critical limitation: their reliance on iterative sampling processes, which often necessitate substantial computational resources and prolonged generation times, particularly for high-resolution

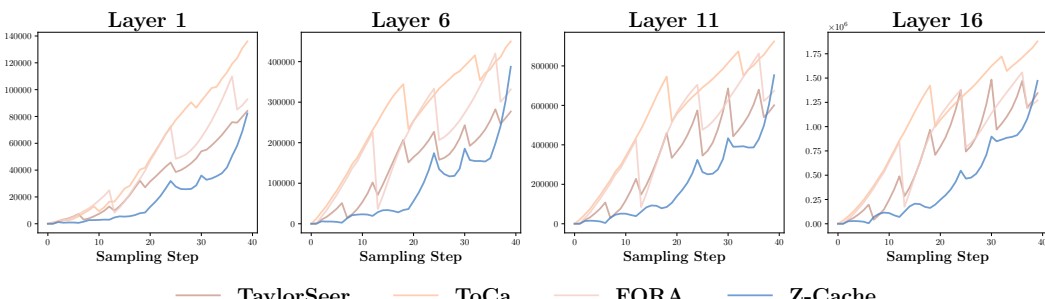

Figure 2: **Visualization of the per-layer L1 error from feature caching.** It is observed that there exists a periodic decline in L1 error, which is caused by the full-computation step in feature caching. The proposed *Z-Cache* utilizes this phenomenon and derives fewer cumulative errors.

outputs. Consequently, the practical deployment of diffusion models remains constrained, hindering their accessibility to broader audiences and applications where efficiency is paramount.

Various acceleration strategies (Selvaraju et al., 2024; Qiu et al., 2025; Zou et al., 2024b; Liu et al., 2024; 2025a; Feng et al., 2025; Zheng et al., 2025; Liu et al., 2025b) have been proposed to mitigate the computational demands of diffusion models. A key observation driving these efforts is the high similarity between hidden states at temporally adjacent steps, which enables caching methodologies to expedite the sampling process. Existing research (Selvaraju et al., 2024; Qiu et al., 2025; Zou et al., 2024b; Liu et al., 2024) has largely focused on identifying reusable intermediate features in both U-Net and transformer-based diffusion architectures, while others (Liu et al., 2025a) have leveraged temporal continuity hypotheses to predict features from preceding sampling steps. However, a critical trade-off persists: as the caching interval widens to reduce computational overhead, the accumulation of errors in predicted or reused features increases proportionally, eventually becoming non-negligible and degrading output quality, which is not acceptable in real-world applications.

To solve this problem, this paper begins with an analysis of the error from feature caching during the generation process, as illustrated in Figure 2. It is observed that the error tends to accumulate in all four feature caching, which is in line with our previous assumption and can explain the loss in generation quality introduced by feature caching. Besides, interestingly, we observe a striking pattern: feature error exhibits a sharp decline precisely at the point of full computation steps, indicating that *the diffusion model can correct itself from a wrong generation trajectory to some extent*. This observation further inspired us to make full use of such self-correction ability. Concretely, we may leverage "future" features, those computed later in the generation process, to reflect on and refine earlier-computed features.

Building on this insight, we introduce *Z-Cache*, a novel caching framework designed to strategically reintegrate current features, enabling periodic correction of initial sampling steps. Central to its operation is a self-correcting mechanism that iteratively refines sampling trajectories, gradually aligning them with the true denoising dynamics. Concretely, Z-Cache is mainly composed of three steps. Firstly, like the previous feature caching methods, Z-Cache performs one-step full computation (*e.g.*, denoted as timestep $t$) and predicts the features in the future caching timesteps (*i.e.,* from $t+1$ to $t+N$). Then, Z-Cache employs the predicted features to skip the computation of diffusion transformer layers in these caching steps, followed by another full computation step (*i.e., $t+N+1$*). After that, instead of moving to the next "computation-then-caching" loop between timestep $t+N+1$ to $t+2N$, Z-Cache **goes back** to the last caching steps ($t+1$ to $t+N$) and updates these predicted values based on an interpolation between the two full-computed steps to perform self-correction. As shown in Figure 2, Z-Cache significantly reduces the caching error by a clear margin. (More analysis on A.1 )

Notably, despite its advanced error-correction capabilities, Z-Cache incurs minimal additional computational overhead, adding just 0.2% FLOPs compared to non-reflection-based approaches. The method derives its name, *Z-Cache*, from its characteristic zig-zag optimization path through the sampling space, a trajectory that balances rigorous error correction with computational efficiency. Extensive experiments on DiT, FLUX.dev, and Hunyuan Video demonstrate its effectiveness on class-to-image generation, text-to-image generation, and text-to-video generation.

## 2 RELATED WORKS

Recent research has shown that diffusion models (Sohl-Dickstein et al., 2015; Ho et al., 2020) excel in generating high-quality images and videos. Early implementations relied heavily on U-Net architectures (Ronneberger et al., 2015), which faced limitations in scaling to larger models. The emergence of Diffusion Transformers (DiT) (Peebles & Xie, 2023b) overcame these constraints, catalyzing innovations that have achieved remarkable results across various applications (Chen et al., 2024b;a; Zheng et al., 2024; Yang et al., 2025). However, the iterative nature of diffusion sampling creates significant computational overhead. Research efforts addressing this challenge fall into two main categories: reducing sampling steps and accelerating the denoising network.

**Sampling Timestep Reduction** Efforts to minimize required sampling iterations while maintaining generation quality have produced several effective strategies. DDIM (Song et al., 2021) pioneered deterministic sampling that preserved quality with fewer steps. Advanced mathematical approaches like DPM-Solver (Lu et al., 2022a;b; Zheng et al., 2023) utilized higher-order ODE solvers to further improve efficiency. Rectified Flow (Liu et al., 2023) established more direct transformation paths between noise and data distributions, while knowledge distillation methods (Salimans & Ho, 2022; Meng et al., 2022) compressed multiple denoising operations. Consistency Models (Song et al., 2023) introduced a breakthrough framework enabling one-step or few-step generation by directly connecting noisy inputs to clean outputs, dramatically improving practical utility.

**Denoising Network Acceleration** Beyond reducing sampling iterations, enhancing the efficiency of the denoising network itself offers complementary performance gains. These approaches divide into compression-based and caching-based techniques. Various compression strategies—including pruning (Fang et al., 2023; Zhu et al., 2024), quantization (Li et al., 2023b; Shang et al., 2023; Kim et al., 2025), distillation (Li et al., 2024), and token reduction (Bolya & Hoffman, 2023; Kim et al., 2024; Zhang et al., 2024; 2025; Cheng et al., 2025; Saghatchian et al., 2025)—reduce computational requirements while preserving output quality. While effective, these approaches typically necessitate additional fine-tuning and carefully balance model size against generative capabilities.

**Feature Caching-based Acceleration** Caching approaches offer training-free acceleration by reusing computational results. Originally conceived for U-Net architectures (Li et al., 2023a; Ma et al., 2024), these techniques have evolved to suit transformer-based models. FasterCache (Lv et al., 2025) implements an efficient cache-then-forecast approach with frequency-selective enhancement of conditional-unconditional residuals. Complementary methods include attention and representation reuse (FORA (Selvaraju et al., 2024), $\Delta$-DiT (Chen et al., 2024c)), timestep-adaptive caching (TeaCache (Liu et al., 2024)), and multi-dimensional attention optimization (DiTFastAttn (Yuan et al., 2024)). The ToCa framework (Zou et al., 2024a;b) preserves information through dynamic updates, while EOC (Qiu et al., 2025) employs error optimization with knowledge extraction. Recent advancements like UniCP (Sun et al., 2025) combine adaptive caching with pruning, and RAS (Liu et al., 2025c) applies spatially-varying sampling rates. TaylorSeer (Liu et al., 2025a) utilizes polynomial extrapolation from neighboring cached activations to predict intermediate features, effectively balancing computational savings with generation quality.

Caching-based methods have evolved with the introduction of the new paradigm "*cache-then-forecast*" (Liu et al., 2025a). However, the inherent non-linearity of DiT leads to an exponential decrease in feature similarity as the time interval increases, posing significant challenges for achieving higher ratio of acceleration. Our work focuses on addressing this issue with the introduction of a new paradigm "*cache-then-check*". In contrast to the single sampling process adopted in previous methods, we adopted two sampling processes. One of these, acting as an auxiliary process, generates potential future feature representations. These are then employed to assess the features in the other process. We integrate the features through a fusion mechanism to obtain more accurate feature representations.

## 3 METHODOLOGY

### 3.1 PRELIMINARY

#### 3.1.1 DIFFUSION MODELS

Diffusion models (Ho et al., 2020; Song et al., 2021) generate structured data by progressively transforming noise into meaningful data through iterative denoising steps. The core mechanism

models the conditional probability distribution at each timestep as a Gaussian. Specifically, the model predicts the mean and variance for $x_{t-1}$ given $x_t$ at timestep $t$. The process is expressed as:

$$p_\theta(x_{t-1}|x_t) = \mathcal{N}\left(x_{t-1}; \frac{1}{\sqrt{\alpha_t}}\left(x_t - \frac{1-\alpha_t}{\sqrt{1-\bar{\alpha}_t}}\tau_\theta(x_t,t)\right), \beta_t\mathbf{I}\right), \tag{1}$$

where $\mathcal{N}$ denotes a normal distribution, $\alpha_t$ and $\beta_t$ are time-dependent parameters, and $\tau_\theta(x_t,t)$ is the predicted noise at timestep $t$. The process begins with a noisy image and iteratively refines it by sampling from these distributions until a clean sample is obtained.

### 3.1.2 DIFFUSION TRANSFORMER ARCHITECTURE

The Diffusion Transformer (DiT) (Peebles & Xie, 2023a) employs a hierarchical structure $\mathcal{G} = g_1 \circ g_2 \circ \cdots \circ g_L$, where each module $g_l = \mathcal{F}_{\mathrm{SA}}^l \circ \mathcal{F}_{\mathrm{CA}}^l \circ \mathcal{F}_{\mathrm{MLP}}^l$ consists of self-attention (SA), cross-attention (CA), and multilayer perceptron (MLP) components. In DiT, these components are dynamically adapted over time to accommodate varying noise levels during the image generation process. The input $\mathbf{x}_t = \{x_i\}_{i=1}^{H \times W}$ is represented as a sequence of tokens corresponding to image patches. Each module integrates information through residual connections, defined as $\mathcal{F}(\mathbf{x}) = \mathbf{x} + \mathrm{AdaLN} \circ f(\mathbf{x})$, where AdaLN refers to adaptive layer normalization, which facilitates more effective learning.

### 3.2 FEATURE CACHING

Recent feature caching methods follow a similar paradigm (Zou et al., 2024b; Lv et al., 2025; Liu et al., 2025a; 2024). Generally, given a typical sampling procedure with timestep scheduler $T$, a diffusion model computes a set of hidden features $\mathcal{H}_t = \{h_t^l\}$ for each layer $l \in L$ at timestep $t$, and a cache strategy $\tilde{\mathcal{H}}_k = \mathcal{C}(\mathcal{H}_A, k)$ takes previously cached features with timesteps $A$ to estimate other hypothesis hidden features $\tilde{\mathcal{H}}_k$ at timestep $k$. Then, a *Naive Feature Caching Strategy*, which periodically reuses computed features for the following $n-1$ timesteps, can be formalized as

$$\tilde{\mathcal{H}}_k = \mathcal{C}(\mathcal{H}_t, k) = \mathcal{H}_t, \text{ for all } t - n < k < t,$$

which can achieve a theoretical $(n-1)$ fold speedup but suffers from cumulative error as $n$ increases.

### 3.3 Z-CACHE

Z-Cache introduces a self-reflection strategy to mitigate predictive errors in conventional cache approaches. It periodically re-visits the past sampling trajectory and updates the intermediate steps with guidance from "future" ones.

**Forward with Cache** During the first stage, *Z-Cache* use a selected base cache strategy to predict intermediate timestep hidden states until the next full computation. That is, given a base cache strategy $\tilde{\mathcal{H}}_k = \mathcal{C}(\mathcal{H}_A, k)$ and a set of cached hidden states $\mathcal{H}_A$, we can estimate all hidden states between timestep interval $(t_1, t_2)$ with

$$\tilde{\mathcal{H}}_k = \mathcal{C}(\mathcal{H}_A, k), \text{ for all } t_1 < k < t_2$$

For base cache strategies like Taylorseer(Liu et al., 2025a) which are predefined with a cache frequency $\mathcal{N}$, the timestep interval becomes $(t - \mathcal{N}, t)$; other heuristic methods like TeaCache (Liu et al., 2024) decide the interval on-the-fly. In this paper, we used TaylorSeer as our base cache strategy and we will discuss the differences between using TeaCache and TaylorSeer in Section 4.5.

**Reflection** After using these estimated hidden states to sample the noised latent from timestep $t$ to $t - n + 1$, we perform a full computation at timestep $t - n$ and update the set of cached hidden states, noted as $\mathcal{H}_{A \cup \{t-n\}}$. At this point, unlike other cache strategies that might continue on following iterations, *Z-Cache* rolls back to the last full activation and re-computes all intermediate timesteps with guidance from the newly computed future hidden states. Formally, given an averaging weight $w$,

$$\tilde{\mathcal{H}}_k = (1-w) \cdot \mathcal{C}(\mathcal{H}_A, k) + w \cdot \mathcal{C}(\mathcal{H}_{A \cup \{t-n\}}, k), \text{ for all } t_1 \leq k < t_2$$

In practice, the averaging weight $w$ is defined as a monotonically decreasing function $w = \mathcal{W}(k)$ with range on $[0,1]$. For example, a logarithmic averaging weight function can be defined as

$$w = \frac{\log(t_2 - k + 1)}{\log(t_2 - t_1 + 1)} \text{ for all } t_1 \leq k < t_2.$$

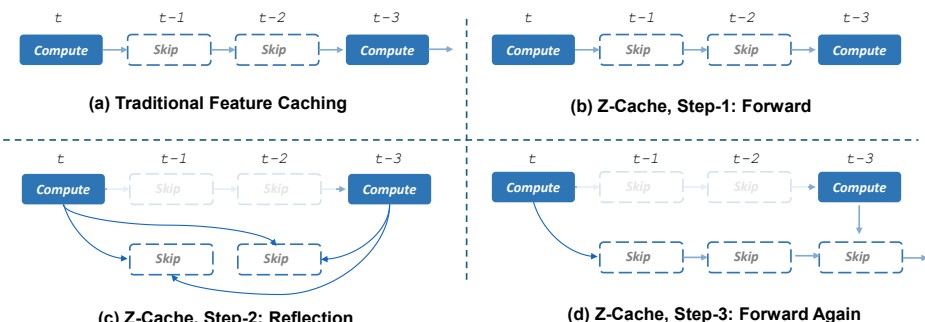

Figure 3: **Visualization of *Z-Cache*'s pipeline**. On the first stage, *Z-Cache* use cached features before timestep $t$ to predict features at $t-1$ and $t-2$, then, it runs a full computation at timestep $t-3$ and cache its features. At second stage, *Z-Cache* uses both cached features at timestep $t$ and $t-3$ separately to re-computed intermediate features of $t-1$, $t-2$, and $t-3$; it then averages both results to perform the re-sampling. At the final stage, *Z-Cache* updates the cached features at timestep $t-3$ with newly averaged features without full computation. Compared with traditional feature caching shown in (a), Z-Cache does not introduce additional full-computation steps, while the computation of caching steps and reflection is ignorable ($\leq 0.1\%$).

For all the following experiments, we chose to use the logarithmic averaging weight function, which has been discussed in Figure 9. Such a definition of the averaging weight makes the process of self-reflection free of hyperparameters. An overview of the pipeline of Z-Cache is shown in Figure 3.

## 4 EXPERIMENTS

### 4.1 EXPERIMENT SETTINGS

We conducted experiments on three state-of-the-art diffusion models: DiT-XL/2 (Peebles & Xie, 2023a), FLUX.1-dev (Labs, 2024), and HunyuanVideo (Li et al.). To maintain implementation parity and ensure fair benchmarking, we restricted our comparative analysis exclusively to caching strategies featuring **officially maintained, model-specific support** for each respective architecture. Therefore, a per-model subset of baselines was systematically selected from the literature: FORA (Selvaraju et al., 2024), ToCa (Qiu et al., 2025), DuCa (Zou et al., 2024b), TeaCache Liu et al. (2024), and TaylorSeer (Liu et al., 2025a). Our evaluation protocol accommodated implementation heterogeneity through a two-tiered approach:

- For methods sharing compatible codebases and standardized evaluation pipelines, we performed comparisons encompassing both high-level perceptual metrics (FID (Heusel et al., 2017), sFID (Ding et al., 2023), Inception Score (Salimans et al., 2016), VBench (Huang et al., 2023)) and low-level pixel-wise metrics (PSNR, SSIM (Wang et al., 2004), LPIPS (Zhang et al., 2018)).

- For techniques requiring substantial adaptation efforts due to incompatible frameworks, evaluation was limited to pixel-level metrics only , with all measurements benchmarked against ground-truth outputs generated from their author-provided implementations.

### 4.2 CLASS-TO-IMAGE GENERATION

For comprehensive evaluation of class-conditional generation on ImageNet (Russakovsky et al., 2015), all experiments employed a DDIM-50 sampling protocol at $256 \times 256$ resolution with 50,000 images generated in total. Methods implementing activation period optimization are explicitly annotated with the operator $\mathcal{N}$. All experiments were executed on NVIDIA H20 GPUs with standardized controls.

Quantitative evaluations on ImageNet class-to-image generation using the DiT-XL/2 model confirm that *Z-Cache* consistently outperforms state-of-the-art caching baselines across key metrics. As Table 1 demonstrates, *Z-Cache* achieves the best performance at all tested acceleration levels. Specifically, at $5.5\times$ acceleration, *Z-Cache* maintains both the lowest FID (3.25) and sFID (5.52) score, and also the highest Inception Score (219.30). On the contrary, FORA, ToCa and DuCa are all

Table 1: **Quantitative comparison on class-to-image generation** on ImageNet with DiT-XL/2. Metrics marked with "-" are left unreported, as they were not provided in the original publications.

| Method | Efficient Attention | Latency(s)↓ | FLOPs(T)↓ | Speed↑ | FID↓ | sFID↓ | Inception Score↑ | PSNR↑ | SSIM↑ | LPIPS↓ |
|---|---|---|---|---|---|---|---|---|---|---|
| **DDIM-50 steps** | ✔ | 0.428 | 23.74 | 1.00× | 2.32 | 4.32 | 241.25 | ∞ | 1.00 | 0.000 |
| **DDIM-12 steps** | ✔ | 0.128 | 5.70 | 4.17× | 7.80 | 8.03 | 184.50 | 22.70 | 0.742 | 0.196 |
| **L2C(NFE=30)** | ✔ | 0.281 | 11.55 | 2.05× | 2.61 | - | 237.83 | - | - | - |
| **SmoothCache**($\alpha = 0.22$) | ✔ | 0.251 | 8.57 | 2.77× | 4.15 | - | 231.71 | - | - | - |
| **DDIM-10 steps** | ✔ | 0.115 | 4.75 | 5.00× | 12.15 | 11.33 | 159.13 | 21.52 | 0.687 | 0.251 |
| **FORA** ($\mathcal{N} = 5$) | ✔ | 0.149 | 5.24 | 4.53× | 6.58 | 11.29 | 193.01 | 21.85 | 0.692 | 0.235 |
| **ToCa** ($\mathcal{N} = 6$) | ✘ | 0.163 | 6.34 | 3.75× | 6.55 | 7.10 | 189.53 | 17.66 | 0.572 | 0.360 |
| **DuCa** ($\mathcal{N} = 5$) | ✔ | 0.154 | 6.27 | 3.78× | 6.06 | 6.72 | 198.46 | 16.65 | 0.543 | 0.390 |
| **TaylorSeer** ($\mathcal{N} = 5$) | ✔ | 0.195 | 5.24 | 4.53× | 2.65 | 5.36 | 231.59 | 28.50 | 0.901 | 0.065 |
| **Z-Cache** ($\mathcal{N} = 5$) | ✔ | 0.185 | 5.25 | 4.53× | 2.46 | 4.71 | 232.34 | 29.24 | 0.908 | 0.052 |
| **FORA** ($\mathcal{N} = 6$) | ✔ | 0.135 | 4.29 | 5.53× | 9.24 | 14.84 | 171.33 | 20.69 | 0.638 | 0.290 |
| **ToCa** ($\mathcal{N} = 9$) | ✘ | 0.127 | 4.54 | 5.23× | 12.86 | 12.82 | 151.37 | 15.71 | 0.485 | 0.473 |
| **DuCa** ($\mathcal{N} = 9$) | ✔ | 0.105 | 4.30 | 5.52× | 12.05 | 11.82 | 156.20 | 15.85 | 0.483 | 0.460 |
| **TaylorSeer** ($\mathcal{N} = 6$) | ✔ | 0.191 | 4.76 | 4.98× | 3.09 | 6.50 | 225.16 | 25.48 | 0.831 | 0.107 |
| **Z-Cache** ($\mathcal{N} = 6$) | ✔ | 0.172 | 4.78 | 4.96× | 2.68 | 4.84 | 228.26 | 27.16 | 0.880 | 0.078 |
| **FORA** ($\mathcal{N} = 7$) | ✔ | 0.123 | 3.82 | 6.22× | 12.55 | 18.63 | 148.44 | 19.87 | 0.598 | 0.335 |
| **ToCa** ($\mathcal{N} = 13$) | ✘ | 0.118 | 4.03 | 5.90× | 21.24 | 19.93 | 116.08 | 15.06 | 0.449 | 0.540 |
| **DuCa** ($\mathcal{N} = 12$) | ✔ | 0.081 | 3.94 | 6.02× | 31.97 | 27.26 | 87.94 | 14.23 | 0.396 | 0.593 |
| **TaylorSeer** ($\mathcal{N} = 7$) | ✔ | 0.177 | 3.82 | 6.22× | 3.60 | 7.07 | 217.79 | 24.09 | 0.794 | 0.136 |
| **Z-Cache** ($\mathcal{N} = 7$) | ✔ | 0.150 | 3.82 | 6.22× | 3.25 | 5.52 | 219.30 | 23.96 | 0.795 | 0.129 |

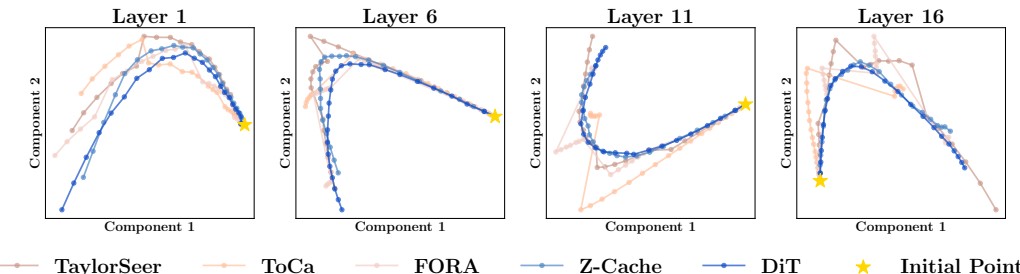

Figure 4: The Trajectories of PCA components of hidden features. *Z-Cache* has the minimum divergence to the ground-truth comparing with other methods.

suffering from catastrophic degradation (FID > 10, sFID > 10, and Inception Score < 200). This signifies *Z-Cache*'s superior preservation of image fidelity and diversity.

Furthermore, as detailed in Table 1, *Z-Cache* achieved a comparable performance with Taylorseer while surpassing other sampling results on the 6.22× acceleration level. This comprehensive performance advantage underscores that *Z-Cache* generates sampling trajectories closely aligned with the original model, minimizing semantic shifts in the output data.

To better illustrate *Z-Cache*'s mechanism, we gathered latent hidden states generated during sampling and compared their trajectories. Visualized in Figure 4, other methods often produce "bumps" during cache time as they do not perceive future states, while *Z-Cache* repetitively reflects past hidden states and therefore lies close to the ground truth trajectories.

### 4.3 Text-to-Image Generation

For text-guided image generation, we maintained model-default sampling configurations throughout all experiments. Each method generated 1,600 high-resolution images (1024 × 1024) using PartiPrompts (Yu et al., 2022) as textual inputs.

On the text-to-image generation task using FLUX.1-dev, *Z-Cache* demonstrates equally compelling advantages. Despite FLUX's distinct architecture and higher-resolution outputs (1024×1024), *Z-Cache* maintains near-original output quality while achieving 5.53× inference acceleration – a critical gain for large-scale deployment. Crucially, this performance aligns with *Z-Cache*'s robustness observed in DiT-XL/2 evaluations, confirming its generalizability across diffusion model families and generative tasks.

We conducted comprehensive visual comparisons across all methods using 10 carefully selected text prompts. These prompts systematically targeted five critical generative challenge domains: stylized illustration, photorealistic scenes, complex culinary subjects, semiotically integrated text, and high-fidelity human anatomical features. Intuitively, these prompts collectively span the most

Table 2: **Quantitative comparison on text-to-image generation.** This table compares the pixel-level metrics on PartiPrompts with FLUX.dev-1.

| Method FLUX.1 | Efficient Attention | Acceleration | | | | PSNR↑ | SSIM↑ | LPIPS↓ |
|---|---|---|---|---|---|---|---|---|
| | | Latency(s)↓ | Speed↑ | FLOPs(T)↓ | Speed↑ | | | |
| [dev]: 50 steps | ✔ | 17.20 | 1.00× | 3719.50 | 1.00× | - | - | - |
| **ToCa** ($\mathcal{N}=9$) | ✘ | 8.59 | 2.00× | 854.42 | 4.35× | 29.02 | 0.6269 | 0.2435 |
| **TeaCache** ($l=0.8$) | ✔ | 6.15 | 2.80× | 891.97 | 4.17× | 29.16 | 0.6985 | 0.2607 |
| **TaylorSeer** ($\mathcal{N}=6$) | ✔ | 7.21 | 2.39× | 820.80 | 4.53× | 28.41 | 0.6736 | 0.1951 |
| **Z-Cache** ($\mathcal{N}=7$) | ✔ | 5.47 | 3.14× | 821.59 | 4.53× | **29.95** | **0.7082** | **0.1773** |
| **ToCa** ($\mathcal{N}=10$) | ✘ | 8.59 | 2.00× | 714.66 | 5.20× | 28.89 | 0.5837 | 0.2924 |
| **TeaCache** ($l=0.99$) | ✔ | 5.30 | 3.25× | 743.63 | 4.89× | 28.92 | 0.6764 | 0.2942 |
| **TaylorSeer** ($\mathcal{N}=7$) | ✔ | 6.77 | 2.54× | 746.28 | 4.98× | 28.35 | 0.6454 | 0.2321 |
| **Z-Cache** ($\mathcal{N}=8$) | ✔ | 5.02 | 3.42× | 747.01 | 4.98× | **29.54** | **0.6908** | **0.2030** |
| **ToCa** ($\mathcal{N}=12$) | ✘ | 8.48 | 2.03× | 649.36 | 5.73× | 28.73 | 0.5556 | 0.3396 |
| **TaylorSeer** ($\mathcal{N}=8$) | ✔ | 6.34 | 2.71× | 671.77 | 5.53× | 28.30 | 0.6197 | 0.2577 |
| **Z-Cache** ($\mathcal{N}=9$) | ✔ | 4.84 | 3.55× | 672.59 | 5.53× | **29.17** | **0.6586** | **0.2320** |

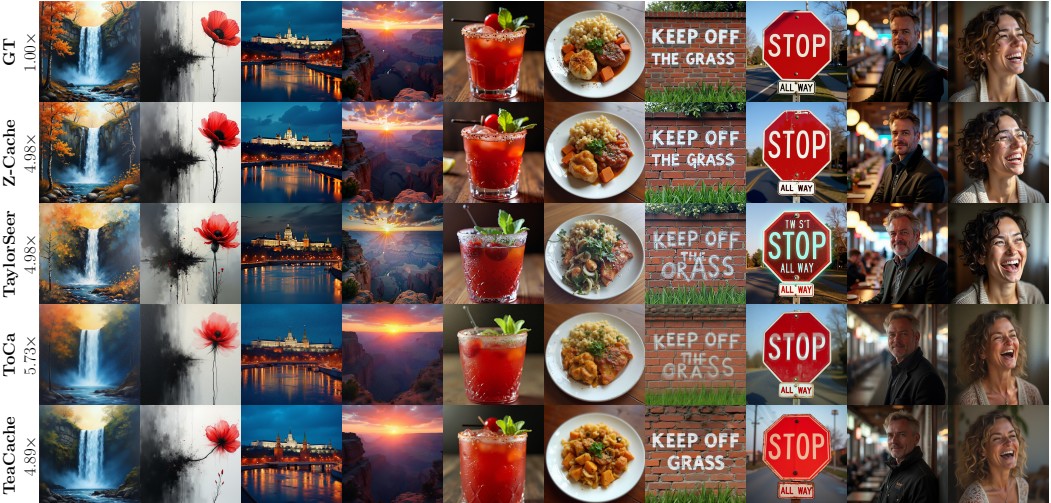

Figure 5: **Visual comparison of 5× accelerated FLUX-1.dev**, demonstrating our method's superior preservation of color fidelity, content integrity, and text legibility over competing approaches.

rigorous stress tests for text-to-image systems. Shown in Figure 5, *Z-Cache* has the smallest semantic changes: Specifically, in the *"Keep off the grass"* image, TaylorSeer and ToCa failed to place the word *"The"* in the right position, and TeaCache ignored the word *"The"* entirely. On the other hand, *Z-Cache* is the only one to correctly write the sentence in the same place as the original one.

We further examine the semantic consistency under progressive acceleration. Figure 6 visualizes output sequences across consecutive $\mathcal{N}$ values from 1 to 9, revealing that *Z-Cache* maintains prompt-aligned content consistency (e.g., stable object counts/positions), whereas TaylorSeer exhibits progressive semantic displacement. Moreover, we compared these two methods on local patches for $\mathcal{N} \geq 7$ in Figure 7: other methods develop systematic distortions, such as hue shifts along high-gradient contours and checkerboard-like artifacts spanning the entire image. These observations corroborate our hypothesis that *Z-Cache*'s dynamic cache reflection mechanism fundamentally mitigates error accumulation in diffusion trajectories.

## 4.4 TEXT-TO-VIDEO GENERATION

For video generation tasks, all experiments utilized the model's default configuration. We generated a total of 4720 video sequences (65 frames per sequence at 640×480 resolution) from VBench with 4 samples each. Evaluation employed the comprehensive VBench framework. Since all methods produce noticeable semantic shifts, we skipped all pixel-level evaluation for the best interpretation.

Extending to video synthesis on HunyuanVideo, *Z-Cache* maintains a consistent and comparable performance with other competing methods on the VBench metric. As table 3 shows, our method

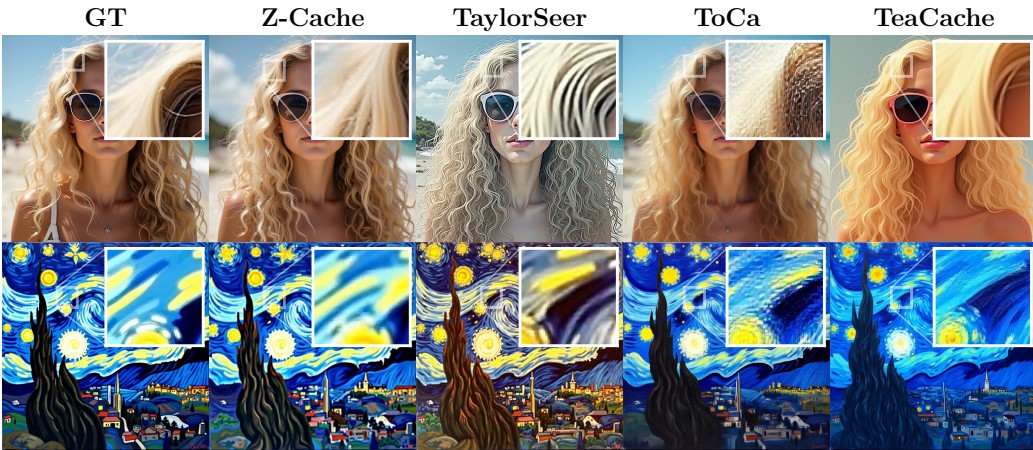

Figure 6: **Visualization of semantic shifts under different caching iterval** $\mathcal{N}$. As the cache interval increases, the image quality of TaylorSeer degrades, whereas Z-Cache maintains the performance, demonstrating the effectiveness of Z-Cache in high-acceleration ratios.

Figure 7: Visualization of local image patches with analogous patterns, comparing two key textural attributes: **fluffy hair (top row)** and **art style (bottom row)**.

achieved the highest VBench scores (80.10 for $4.96\times$ acceleration and 79.60 for $6.20\times$ acceleration) across different acceleration levels. This demonstrates *Z-Cache*'s advantage for video diffusion acceleration, where existing caching strategies struggle to balance efficiency with temporal consistency.

Beyond quantitative metrics, we present qualitative comparisons of these methods on identical frame sequences in Figure 8. For instance, in the "a motorcycle" video, Z-Cache generates a coherent depiction of the rider's face, whereas methods like TaylorSeer produce distorted facial features or fragmented figures. In the *"a person is dunking a basketball"* video, approaches such as ToCa entirely omit the basketball, failing to render it altogether. Finally, in the *"a fork on the right of a knife"* video, all baseline methods either produce malformed or misshapen knives or completely disregard the fork, whereas Z-Cache accurately captures both objects in their correct spatial arrangement.

### 4.5 ABLATION STUDY

Table 4: **Ablation study** on a $6.22\times$ accelerated FLUX-dev.

| Method | SSIM↑ | LPIPS↓ |
|---|---|---|
| **TaylorSeer** ($\mathcal{N} = 8$) | 0.5905 | 0.3084 |
| **Z-Cache (TaylorSeer, $\mathcal{N} = 8$)** | 0.6908 | 0.2030 |
| **TeaCache** ($l = 0.99$) | 0.6298 | 0.3107 |
| **Z-Cache (TeaCache, $l = 0.99$)** | 0.6374 | 0.3113 |

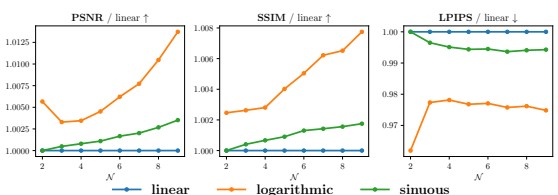

Figure 9: Performance of different averaging functions on FLUX.dev-1. All results were divided by their linear version for clarity.

**Base Cache Strategy** *Z-Cache* is naturally orthogonal to all single-trajectory cache methods, so it is possible to choose a different base cache strategy besides TaylorSeer. Here we compared

Table 3: **Quantitative comparison on text-to-video** generation with Hunyuan Video on VBench.

| Method (on HunyuanVideo) | Efficient Attention | Acceleration | | | | VBench ↑ Score(%) |
|---|---|---|---|---|---|---|
| | | Latency(s) ↓ | Speed ↑ | FLOPs(T) ↓ | Speed ↑ | |
| **Original: 50 steps** | ✔ | 145.00 | 1.00× | 29773.0 | 1.00× | 80.66 |
| **22% steps** | ✔ | 31.87 | 4.55× | 6550.1 | 4.55× | 78.74 |
| **TeaCache**($l = 0.4$) | ✔ | 26.61 | 5.45× | 6550.1 | 4.55× | 79.36 |
| **FORA**($N = 5$) | ✔ | 34.39 | 4.22× | 5960.4 | 5.00× | 78.83 |
| **ToCa** ($\mathcal{N} = 5, R = 90\%$) | ✘ | 38.52 | 3.76× | 7006.2 | 4.25× | 78.86 |
| **DuCa** ($\mathcal{N} = 5, R = 90\%$) | ✔ | 31.69 | 4.58× | 6483.2 | 4.48× | 78.72 |
| **TaylorSeer** ($\mathcal{N} = 5, O = 1$) | ✔ | 34.84 | 4.16× | 5960.4 | 5.00× | 79.93 |
| **Z-Cache** ($\mathcal{N} = 5, O = 1$) | ✔ | 28.69 | 5.05× | **5997.2** | 4.96× | **80.10** |
| **TeaCache**($l = 0.5$) | ✔ | 26.61 | 5.45× | 5359.1 | 5.56× | 78.32 |
| **TaylorSeer** ($\mathcal{N} = 7, O = 1$) | ✔ | 28.82 | 5.03× | 4794.5 | 6.21× | 79.09 |
| **Z-Cache** ($\mathcal{N} = 7, O = 1$) | ✔ | 23.08 | 6.28× | **4799.6** | 6.20× | 79.60 |

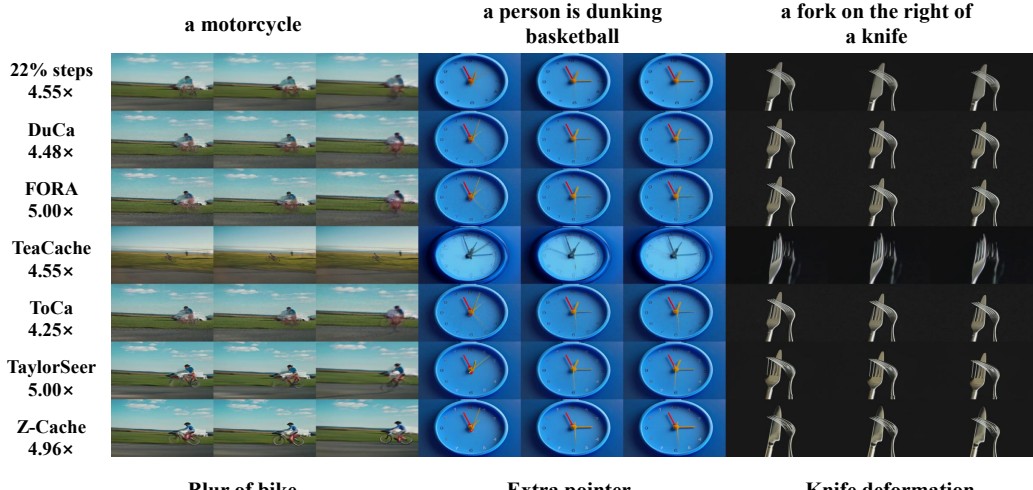

Figure 8: Visualization of videos generated by Hunyuan Video with different feature caching methods.

different strategies with the same setting we did on FLUX.1-dev, shown in table 4. Under different acceleration levels, TaylorSeer provides the optimal performance, and we therefore used it as our cache predictor. When applying *Z-Cache* to TeaCache, it still brings better generation quality under the same acceleration ratios, demonstrating its generalization ability.

**Averaging Weight Function** Moreover, we compared different averaging weight functions over consecutive increasing $\mathcal{N}$. The results in Figure 9 clearly show the advantage of using a logarithmic averaging weight function (as discussed in Section 3.3), which leads to consistently better performance, indicating that the averaging weight function does not need to introduce an additional hyperparameter for balance.

## 5 CONCLUSION

This paper introduces *Z-Cache*, a novel caching framework that redefines the efficiency-quality trade-off for diffusion transformers. Central to its innovation is a self-reflective mechanism, which periodically refines intermediate features through "future-aware" feature integration—enabling substantial acceleration without sacrificing output fidelity. By integrating this mechanism, *Z-Cache* achieves exceptional acceleration (up to 5.53× faster inference on FLUX-dev.1) while preserving high output quality. Beyond static image generation, Z-Cache demonstrates remarkable adaptability to video tasks, attaining a state-of-the-art 6.20× acceleration rate for videos, all while maintaining temporal consistency and high output quality.

# A APPENDIX

## A.1 THEORATICAL ANALYSIS OF MOTIVATION

Our design is motivated by a **smoothness (Lipschitz)** hypothesis on the denoising network's hidden states with respect to the timestep. Let the diffusion timesteps be

$$t_0 > t_1 > \cdots > t_T$$

and the hidden feature denote by $H_\ell(t_i) \in \mathbb{R}^d$ at layer $\ell$ and timestep $t_i$ when running the full (non-accelerated) model. We assume that for each layer $\ell$, the mapping $t \mapsto H_\ell(t)$ is **Lipschitz-continuous** with constant $L_\ell$:

$$\left| H_\ell(t_i) - H_\ell(t_j) \right| \leq L_\ell, |t_i - t_j| \quad \forall i, j$$

Consider a cache window $t_a, t_b$ with $t_a > t_b$, where we compute full features at the endpoints $t_a$ and $t_b$ and cache intermediate steps $k \in (a, b)$. Classical cache-and-forecast methods (e.g., TaylorSeer) construct an approximation for an intermediate step $k$ **using only the past anchor** at $t_a$:

$$\hat{H}_\ell^{\text{past}}(t_k) \approx H_\ell(t_k) \quad \text{based on } H_\ell(t_a)$$

Under the Lipschitz assumption, this necessarily involves a form of **extrapolation** over a temporal distance $|t_k - t_a|$, so the approximation error can accumulate as we move away from $t_a$.

In Z-Cache, once we reach the next full step at $t_b$, we obtain another anchor $H_\ell(t_b)$ and form **two predictions** for each intermediate step:

$$\hat{H}_\ell^{\text{past}}(t_k) \quad \text{and} \quad \hat{H}_\ell^{\text{future}}(t_k)$$

where $\hat{H}_\ell^{\text{future}}(t_k)$ is obtained by re-running the cache rule with the extended cache that includes $t_b$. We then define the reflected feature as a convex combination

$$\tilde{H}_\ell(t_k) = (1 - w_k)\hat{H}_\ell^{\text{past}}(t_k) + w_k \hat{H}_\ell^{\text{future}}(t_k)$$

with a distance-dependent weight

$$w_k = \varphi\left(\frac{t_a - t_k}{t_a - t_b}\right), \quad \varphi : [0, 1] \to [0, 1] \text{ monotone decreasing}$$

Let the ground-truth error at step $t_k$ for the two predictors be

$$e_k^{\text{past}} = H_\ell(t_k) - \hat{H}_\ell^{\text{past}}(t_k), \qquad e_k^{\text{future}} = H_\ell(t_k) - \hat{H}_\ell^{\text{future}}(t_k)$$

By linearity and the triangle inequality, the error of Z-Cache's reflected feature satisfies

$$||H_\ell(t_k) - \tilde{H}_\ell(t_k)|| = ||(1 - w_k)e_k^{\text{past}} + w_k e_k^{\text{future}}|| \leq (1 - w_k)||e_k^{\text{past}}|| + w_k||e_k^{\text{future}}||$$

Thus the Z-Cache error at each step is bounded by a convex average of the two single-anchor errors. Under the smoothness hypothesis, and because $t_a$ and $t_b$ are full (low-error) projections back to the true trajectory, we typically have $||e_k^{\text{past}}||$ small near $t_a, ||e_k^{\text{future}}||$ small near, $t_b$ and both moderate in the middle of the window. Choosing $w_k$ to increase with the normalized distance from $t_a$ naturally shifts trust from the past predictor to the future predictor as we move through the window, which reduces the average error compared to using either predictor alone across the whole interval.

Intuitively, this is exactly what one would expect from a smooth trajectory: for any $t_k \in (t_a, t_b)$, the ideal hidden state $H_\ell(t_k)$ should lie close to a suitable convex combination of the two anchors $H_\ell(t_a)$ and $H_\ell(t_b)$. Z-Cache approximates this behavior at the level of cached predictors rather than raw features. In practice, this manifests as:

- **Tighter feature trajectories** in PCA space, where Z-Cache remains closer to the full DiT trajectory than single-anchor cache baselines
- **Better semantic faithfulness**, as reflected in our quantitative results: Z-Cache consistently improves CLIPScore and ImageReward over TaylorSeer at the same acceleration ratio, while also improving or matching PSNR/SSIM/LPIPS.

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
