# OpenReview forum: "Z-Cache: Accelerating Diffusion Transformers via  Self-Reflection"
_ICLR.cc/2026/Conference — Submitted to ICLR 2026_

### Official Review · Reviewer_aZpU · 2025-10-28

**Soundness:** 2
**Presentation:** 2
**Contribution:** 2
**Rating:** 4
**Confidence:** 4

**Summary:**

This paper proposes z-cache, a solution to reduce the computational cost during the denoising process of diffusion models by caching features, thereby improving inference speed. The difference from previous cache-based acceleration methods is that this paper finds the error accumulation caused by feature caching can be alleviated after a full computation, thus considering that diffusion models can self-correct. The paper suggests using the feature after a future full computation to correct the previous cached feature, mitigating the accumulated errors. This paper conducts extensive comparisons and experiments with various baseline methods on tasks such as c2i, t2i, and t2v.

**Strengths:**

1. The motivation of the paper is clear, and fig.2 demonstrates the effect of error accumulation and the corrective impact brought by full computation.
2. The experimental results are abundant, with extensive comparisons made across various tasks such as t2i, c2i, t2v, and different baseline solutions.
3. The paper is well-written and easy to read.

**Weaknesses:**

1. There are no quantitative metrics for image quality and text consistency in t2i tasks, only some low-level metrics like PSNR reflect similarity to the original image. If fair comparisons with other methods are difficult to reproduce, at least a comparison with the direct baseline method TaylorSeer is necessary.
2. The error introduced by feature caching may have little impact on image generation tasks without reference images, but for some image editing tasks that require consistency in non-edited regions and personalized generation that maintains IP consistency, could the error from feature caching potentially harm critical visual information needed to discern the identity of the original subject or background?
3. I have some doubts about the motivation of the article. It is reasonable that the paper observes that full computation can correct the cumulative errors caused by feature caching at the current time step, but why is the feature at future time steps, when weighted with cached feature from previous time steps, closer to the ground truth feature of the midstep time steps?
4. The novelty of the paper is limited, and the technical contribution is incremental. Using the feature computed fully at future time steps to correct the cached representation at previous time steps seems merely a reverse thinking of TaylorSeer, which uses representations from past time steps to correct future feature, this paper offering limited technical contributions.

**Questions:**

Why is the weighting function $ w( \cdot ) $ designed in the form of a monotonically decreasing logarithmic function, and what is the motivation behind it?

---

> ### Author Response · Authors · 2025-11-22
>
> **Page 1/3**
>
> We thank the reviewer for the thoughtful assessment and helpful suggestions.
>
> **1. Quantitative metrics for t2i and comparison with TaylorSeer**
>
> *We recently finished implementing Z-Cache with diffusers, it allows us to fairly compare with other diffusers-based methods.*
>
> For FLUX.1-dev text-to-image, our goal is to measure how closely an accelerated sampler tracks the original (full-step) model, both in low-level fidelity and semantics. Following your suggestion, we additionally evaluated **CLIPScore** and **ImageReward**. On FLUX.1-dev, for the acceleration settings we obtain:
>
> |                   | CLIPScore  | ImageReward |
> | ----------------- | ---------- | ----------- |
> | TaylorSeer (n=8)  | 32.587     | 0.929       |
> | **Z-Cache (n=8)** | **32.722** | **0.972**   |
> | TaylorSeer (n=7)  | 32.481     | 0.970       |
> | **Z-Cache (n=7)** | **32.774** | **0.978**   |
>
> Across all tested settings, Z-Cache consistently improves both CLIPScore and ImageReward over TaylorSeer at the **same acceleration ratio**, indicating better semantic alignment with the text prompts and higher predicted human preference.
>
> **2. Impact on image editing / personalized generation (IP consistency)**
>
> To directly check this, we evaluated Z-Cache on an image-editing setup instead of only free-generation. Concretely, we ran **FLUX-Kontext-dev** on the **Gedit-bench** image editing benchmark and compared TaylorSeer and Z-Cache:
>
> | Method    | PSNR   | SSIM   | LPIPS  |
> | --------- | ------ | ------ | ------ |
> | TaylorSeer(n=8)   | 29.014 | 0.4338 | 0.4448 |
> | **Z-Cache(n=8)** | **29.511** | **0.5114** | **0.3694** |
> | TaylorSeer(n=7)  | 29.675 | 0.5013 | 0.3658 |
> | **Z-Cache(n=7)** | **30.279** | **0.5667** | **0.3015** |
>
> Compared with TaylorSeer, Z-Cache improves PSNR and SSIM while substantially reducing LPIPS at the same acceleration ratio. Higher PSNR/SSIM and lower LPIPS indicate that Z-Cache better preserves the original image structure and perceptual details, which are precisely the signals needed to retain the identity of the subject and non-edited background regions.

---

> ### Author Response · Authors · 2025-11-26
>
> **Page 2/3**
>
> **3. Why do future features help correct mid-step features?**
>
> In the revised paper we added **Appendix A.1** to formalize this intuition via a simple **smoothness (Lipschitz) hypothesis** on the hidden states of the full (non‑accelerated) sampler. Here we summarize the key points:
>
> - Let $H_\ell(t)$ be the hidden feature at layer $\ell$ and timestep $t$ when we run the *full* model (no caching). Empirically, Fig. 2 shows that within a cache window the L1 error of cached features grows gradually, and then **drops sharply** at every full step. This strongly suggests that the full model trajectory is relatively smooth in $t$, and full steps “re‑project” the state back to a low‑error manifold.
>
> - Assume for each layer $\ell$, the mapping $t \mapsto H_\ell(t)$ is **Lipschitz‑continuous** with respect to the timestep. Intuitively, this means that the true trajectory does not oscillate wildly between two consecutive full steps; it changes gradually.
>
> Consider a cache window bounded by two consecutive full timesteps $t_a > t_b$. Classical cache‑then‑forecast methods (e.g., TaylorSeer) approximate intermediate steps $t_k \in (t_a, t_b)$ using **only the past anchor** at $t_a$. Under the smoothness assumption, this is a form of **extrapolation** across the entire distance $|t_k - t_a|$, so the approximation error tends to accumulate as we move away from $t_a$.
>
> In **Z‑Cache**, once we compute the next full step at $t_b$, we obtain a second anchor and:
>
> 1. Build a “past‑only” prediction $\hat{H}^{\text{past}}(t_k)$ (what classical cache would give),
>
> 2. Build a “future‑aware” prediction $\hat{H}^{\text{future}}(t_k)$ using the extended cache that includes $t_b$, and
>
> 3. Form the **reflected** feature as a convex combination
>   $\tilde{H}(t_k) = (1 - w_k)\hat{H}^{\text{past}}(t_k) + w_k \hat{H}^{\text{future}}(t_k)$
>
>   where $w_k$ increases with the distance from $t_a$ to $t_k$.
>
> Let $e_k^{\text{past}}$ and $e_k^{\text{future}}$ be the prediction errors of the two single‑anchor estimates. A simple triangle‑inequality argument (given in Appendix A.1) shows that the error of Z‑Cache satisfies
> $|H(t_k) - \tilde{H}(t_k)|
> \le (1 - w_k)|e_k^{\text{past}}| + w_k |e_k^{\text{future}}|$
>
> Thus the Z‑Cache error at each step is bounded by a **convex average** of the two single‑anchor errors. Under the smoothness hypothesis and the fact that both endpoints $t_a, t_b$ are full, low‑error projections:
>
> - $|e_k^{\text{past}}|$ is small near $t_a$,
>
> - $|e_k^{\text{future}}|$ is small near $t_b$,
>
> so the convex bound is typically **lower** than using either predictor in isolation across the whole window.
>
> Intuitively: for any intermediate timestep, the *ideal* hidden state should lie close to an appropriate convex combination of the two full‑step anchors. Z‑Cache approximates exactly this behavior at the level of cached predictors instead of raw features.
>
> Empirically, this is reflected in:
>
> - **Fig. 2**, where the per‑layer L1 error of Z‑Cache is lower than TaylorSeer/TeaCache/FORA at the same cache interval; and
>
> - **Fig. 4**, where the PCA trajectories of Z‑Cache stay closest to the full DiT trajectory, while other cache methods show visible “bumps” inside cache windows.
>
> We will explicitly point to Appendix A.1 from Sec. 3.3 so that this connection is clear.

---

> ### Author Response · Authors · 2025-11-26
>
> **Page 3/3**
>
> **4. Novelty beyond “reverse TaylorSeer”**
>
> We understand the concern that “using future features to correct past ones” could sound like just the reverse of TaylorSeer. We see the novelty of Z‑Cache as lying in **how** these features are used and in the overall **cache‑then‑check** sampling pattern:
>
> 1. **Dual‑anchor reflection vs. single‑trajectory extrapolation.**
>   Classical cache‑then‑forecast methods commit to a *single* extrapolated trajectory through each cache window, anchored at a past full step. Z‑Cache instead constructs **two anchored predictions** (past and future) and defines a reflected state whose error is provably bounded by a convex combination of their individual errors (Appendix A.1). This is not simply “using the future” but a principled **two‑sided correction** of the trajectory.
>
> 2. **Trajectory structure: “cache‑then‑check” zig‑zag.**
>   Algorithmically, Z‑Cache changes the **shape** of the sampling trajectory: first we run a standard cache‑then‑forecast rollout, then after the next full step we **roll back and resample** all cached timesteps using the fused features, and only then continue forward. This “zig‑zag” pattern is qualitatively different from any single‑pass scheme (including a naive “reverse TaylorSeer”), and is what produces the periodic error drop in Fig. 2 and the tight PCA trajectories in Fig. 4.
>
> 3. **Orthogonality to base cache rules and architectures.**
>   Z‑Cache is a **framework** that can wrap any single‑trajectory cache baseline. In the rebuttal we showed that, at matched acceleration ratios, Z‑Cache consistently improves TaylorSeer and TeaCache on DiT‑XL/2, FLUX‑dev, and HunyuanVideo, and also on a **U‑Net‑based SDXL** model (where Z‑Cache improves PSNR/SSIM/LPIPS and ImageReward over TaylorSeer at 2.6× and 4.0× speed‑ups). In addition, combining Z‑Cache with **8‑bit quantization** further improves quality over 8‑bit+TaylorSeer at the same speed, showing that our reflection is complementary to compression‑based accelerations.
>
>
> Taken together, the dual‑anchor reflection, changed trajectory structure, cross‑architecture improvements (DiT + U‑Net), and complementarity with quantization methods make Z‑Cache more than a simple “reverse TaylorSeer” variant.
>
> **5. Why a monotonically decreasing logarithmic weighting function?**
>
> Finally, regarding the design of the weighting function:
>
> $\tilde{H}(t_k) = (1 - w_k)\hat{H}^{\text{past}}(t_k) + w_k \hat{H}^{\text{future}}(t_k)$
>
> we use a **monotone logarithmic schedule** $w_k = \varphi(k)$ that depends only on the normalized position of $t_k$ between the two full steps.
>
> Our design choices are:
>
> 1. **Hyperparameter‑free, plug‑and‑play.**
>   We intentionally avoid introducing new tunable scalars (e.g., temperature, slope, bias). The log schedule is fully determined by the window endpoints and the discrete timestep index, so Z‑Cache can be dropped into an existing cache method without any extra tuning effort.
>
> 2. **Matches the smoothness‑based intuition.**
>   Under the Lipschitz view above, the past‑only predictor is more accurate near the earlier endpoint, and the future‑aware predictor is more accurate near the later endpoint. Intuitively，a smooth function such as linear or sinusoidal should work, and we add another logarithmic schedule which **slightly weighted more on the future endpoint** for the reason that we believe the future anchor might contains more semantic information of clear images. **It is also a suprise to us that the logarithmic one works the best on almost all cases**.
>
> 3. **Empirical ablation.**
>   In Fig. 9 of the revised paper we compare **linear**, **logarithmic**, and **sinusoidal** schedules on FLUX‑dev. The logarithmic weighting consistently yields slightly better PSNR, SSIM, and LPIPS than the other two, while remaining parameter‑free. This suggests that within the family of simple monotone functions, the log form best matches the underlying error profile without adding complexity.

---

> ### Author Response · Authors · 2025-11-27
>
> Dear Reviewer aZpU,
>
> Thank you again for your helpful comments, which have been very valuable for improving our paper. In our previous response, we have revised the manuscript and added clarifications and experiments according to your suggestions. If you have any further questions or comments, please feel free to let us know, we would be very happy to continue the discussion.

---

### Official Review · Reviewer_3euh · 2025-10-28

**Soundness:** 3
**Presentation:** 4
**Contribution:** 3
**Rating:** 6
**Confidence:** 5

**Summary:**

The paper presents Z-Cache, a training-free acceleration framework for diffusion transformers that revisits and refines cached features within the sampling trajectory. By introducing a self-reflection mechanism, Z-Cache slashes cumulative error. Extensive evaluations on ImageNet-1K, PartiPrompts and VBench show that DiT, FLUX and HunyuanVideo all enjoy up to 5.53x-6.22× speed-up while retaining top-tier FID, IS, LPIPS and VBench score

**Strengths:**

1.	It is interesting to observe that the feature error drops sharply exactly at the point where full computation steps are completed.
2.	The manuscript is well-structured, clearly written, and easy to follow.

**Weaknesses:**

1.	What operation does the circle symbol in lines 172 and 173 represent?
2.	It would be great if the proposed method could further improve the sampling speed of the distilled models.
3.	It's better to provide a user study to verify, through human evaluation, whether the generative performance of the method is close to the baseline.
4.	The layout of the paper is somewhat inconsistent. Most table captions are placed above the tables, while the caption of Table 4 is below the table.

**Questions:**

The use of future features to update the cache is an interesting idea, and the method demonstrates its ability to accelerate multi-step sampling. It would be valuable to further examine whether the approach is effective for few-step distilled mode

---

> ### Author Response · Authors · 2025-11-21
>
> We thank the reviewer for the positive assessment and helpful suggestions.
>
> **1. Meaning of the circle symbol in lines 172–173**
>
> The circle “$\circ$” denotes function composition. In Sec. 3.1.2 we write the DiT as $\mathcal{G} = g_1 \circ g_2 \circ \cdots \circ g_L$, meaning that the input is processed by block
> $g_1$, then $g_2$, and so on, exactly as in a standard stacked transformer. We will add a short explanation to the text to avoid confusion.
>
> **2. Applicability to distilled / few-step models**
>
> We agree that evaluating Z-Cache on few-step distilled models (e.g., consistency-distilled or DPM-distilled backbones) is interesting. Conceptually, the reflection mechanism remains valid because these models still have multiple denoising steps and internal layers where feature caching can be applied.
>
> To verify this, we tested Z-Cache, TaylorSeer and ToCa on Qwen-Image-Lightning (8-step). The results are:
>
>
> | Method     | PSNR↑  | SSIM↑  | LPIPS↓ |
> | ---------- | ------ | ------ | ------ |
> | Z-Cache (n=2)   | **30.440** | **0.6785** | **0.2910** |
> | TaylorSeer (n=2) | 30.222 | 0.6734 | 0.2984 |
> | ToCa (n=2) | 29.546 |  0.6780  |  0.2938  |
>
> Importantly, even in this few-step regime, Z-Cache still achieve much better performance than ToCa and TaylorSeer, indicating that cache-based acceleration remains effective on distilled few-step models.
>
> **3. User study**
>
> In addition to automatic metrics, we conducted a small-scale (recieved 30 valid responses) user study directly comparing Z-Cache and TaylorSeer. We generated 16 pairs of images (one from Z-Cache and one from TaylorSeer for each prompt) and asked annotators to choose which image they preferred. Overall, **74.5%** of the selections favored **Z-Cache**, and at the prompt level Z-Cache was preferred in **15 out of 16** cases.
>
> These results align with our quantitative metrics (PSNR/SSIM/LPIPS, as well as CLIP-based measures reported elsewhere in the rebuttal) and provide human-evaluation evidence that Z-Cache produces outputs that are more preferred than those of TaylorSeer at the same acceleration ratio. We will briefly report this user study and its outcome in the revised version.
>
> **4. Paper layout**
>
> We will fix the caption placement inconsistency (Table 4 and any others) and carefully re-check the formatting of the final version.

---

> ### Author Response · Authors · 2025-11-27
>
> Dear Reviewer 3euh,
>
> Thank you again for your helpful comments, which have been very valuable for improving our paper. In our previous response, we have revised the manuscript and added clarifications and experiments according to your suggestions. If you have any further questions or comments, please feel free to let us know, we would be very happy to continue the discussion.

---

### Official Review · Reviewer_bgiZ · 2025-10-31

**Soundness:** 3
**Presentation:** 3
**Contribution:** 3
**Rating:** 4
**Confidence:** 4

**Summary:**

To solve the issue of large computation costs during generation with diffusion transformers, this paper proposes a new approach named “Z-cache”. The Z-cache is designed to predict the features in the future caching steps and then perform a full computation. Experiments show the faster inference speed for t2i generation.

**Strengths:**

1.	The proposed method belongs to training-free paradigm, which makes it easy to use.
2.	The results of FID, SSIM, LPIPS demonstrate the effectiveness of the proposed method.
3.	The writing is easy to understand, and the painting is well-drawn.

**Weaknesses:**

1.	In the paper, authors indicate that “the diffusion model can correct itself from a wrong generation trajectory to some extent” and this is the key idea to make full use of self-correction ability. However, there is no further theoretical analysis or explanation to answer why the self-reflection mechanism reduces error. The mechanism behind the observation is important for deep research of DiTs and the robustness of the model.
2.	The Z-cache only repairs the hidden states in the network, instead of the semantic consistency, so, if model generates some wrong semantic contents in i-th step, it is still maintained in future steps.
3.	Whether the proposed Z-cache is still useful under the “residual connection” structures?
4.	Limited experiments:
a)	The paper primarily compares Z-cache with other caching-based methods, however, it lacks a comprehensive comparison with non-caching acceleration techniques, including quantization, pruning, or knowledge distillation.
b)	The experiments are mainly tested on the DiT-based methods, I wonder whether the method is also effective on U-Net-based frameworks?
c)	Hyperparameters: although the paper claims the logarithmic averaging weight eliminates hyperparameter tuning, the choice of base caching strategy (e.g., TaylorSeer vs. TeaCache) still impacts performance.
d)	In Fig. 5, some cases do not support the superiority of the proposed method compared with other models, such as TaylorSeer. This includes the cases in the first, second, third, fourth, and fifth rows.
5.	The failure cases and more analyses should be discussed.

**Questions:**

See above.

---

> ### Author Response · Authors · 2025-11-23
>
> **Page 1/2**
>
> We appreciate your detailed review and insightful questions. Below we address each point and summarize the changes and new experiments we have added.
>
>
> **1. Why does the self-reflection mechanism reduce error?**
>
> In the revised version we added a short theoretical motivation in **Appendix A.1**, formalizing the “self-correction” intuition.   The key assumption is a **smoothness / Lipschitz hypothesis** on the hidden states with respect to timestep:
>
> * Let $H_\ell(t)$ be the hidden feature at layer $\ell$ and timestep $t$ along the **full** (non-accelerated) trajectory.
> * We assume $H_\ell(t)$ changes smoothly in $t$ (Lipschitz continuity), which is consistent with our empirical observation that the L1 error of cached features grows gradually within a cache window and **drops sharply** when a full step is executed (Fig. 2).
>
> In a cache window $[t_a, t_b]$ (two consecutive full steps), classical cache‑then‑forecast methods (e.g., TaylorSeer) approximate intermediate steps using **only the past anchor** at $t_a$, effectively extrapolating over a temporal distance $|t_k - t_a|$. In Z‑Cache, once we compute the next full step at $t_b$, we obtain a **second anchor**, and for every intermediate step $t_k$ we construct:
>
> * a “past-only” prediction $\hat H^{\text{past}}(t_k)$ from ${t_a}$, and
> * a “future-aware” prediction $\hat H^{\text{future}}(t_k)$ from ${t_a, t_b}$.
>
> The reflected feature is their convex combination
>
> $\tilde H(t_k) = (1-w_k)\hat H^{\text{past}}(t_k) + w_k\hat H^{\text{future}}(t_k),$
>
> with a **distance-dependent weight** $w_k$ that increases when moving from $t_a$ to $t_b$. Under the smoothness assumption, the error of $\tilde H(t_k)$ is upper‑bounded by a **convex average** of the two single‑anchor errors (Appendix A.1), so it is typically smaller than using only one anchor over the whole window.
>
> Empirically, this is exactly what we observe:
>
> * In **Fig. 2**, the per-layer L1 error curve of Z‑Cache is consistently lower than that of TaylorSeer and other cache baselines at the same cache interval.
> * In **Fig. 4**, PCA trajectories of Z‑Cache stay closest to the full DiT trajectory, while other methods show noticeable “bumps” during cache segments.
>
> We will explicitly point the reader to Appendix A.1 in Sec. 3.3 of the revised paper and briefly summarize this intuition there.
>
> **2. Does Z‑Cache only repair hidden states but not semantic consistency?**
>
> We agree that semantic consistency is crucial. Mechanically, Z‑Cache **does more than just adjust internal activations**:
>
> * During the reflection stage, we **discard the previously sampled latents** inside the cache window.
> * We recompute all hidden states using the fused (past+future) cache, and then **re-run the denoising network** at each intermediate timestep.
> * The latents ${x_t}$ in that window are therefore **fully resampled** using corrected features, so semantic content can be changed, not merely fine‑tuned.
>
> This effect is visible in several places in the paper:
>
> * In **Fig. 5 (“Keep off the grass”)**, only Z‑Cache preserves the full sentence (“Keep off the grass” including “The”) at the correct location, whereas TaylorSeer and ToCa misplace or drop words, and TeaCache omits “The”.
> * In **Fig. 7**, Z‑Cache better preserves local textures (hair and art style) compared to other cache methods under the same acceleration ratio, indicating improved semantic and structural fidelity rather than just smoother activations.
>
> We will add an explicit sentence in Sec. 3.3 clarifying that Z‑Cache **resamples the intermediate latents**, not just hidden states.
>
>
> **3. Usefulness under residual connection structures**
>
> All three backbones we evaluate—**DiT‑XL/2, FLUX.1‑dev, and HunyuanVideo**—are transformer-based or hybrid architectures with standard residual connections of the form
> $x_{l+1} = x_l + f_l(x_l, t, c)$.
>
> Z‑Cache *only* modifies how the branch $f_l(\cdot)$ is computed at cached steps (via cache prediction and reflection). The residual skip $x_l$ remains unchanged. The consistent improvements we observe across all three residual backbones therefore directly demonstrate that Z‑Cache is **compatible with and useful under residual architectures**.
>
> If there are other specific residual variants the reviewer has in mind (e.g., cross‑layer or multi‑scale residuals in specialized U‑Nets), Z‑Cache can be applied in exactly the same way to the residual branch, and we are happy to explore these in future work.

---

> ### Author Response · Authors · 2025-11-23
>
> **Page 2/2**
>
> **4. “Limited experiments”: non‑caching accelerations, U‑Net models**
>
> *4a. Comparison with non‑caching acceleration methods (quantization / knowledge distillation)*
>
> We extended our experiments to compare directly with quantization‑ and architecture‑based accelerations, and also to **combine** Z‑Cache with 8‑bit quantization. On FLUX.1‑dev we obtain:
>
> | Method               | PSNR       | SSIM   | LPIPS      | ImageReward | Acceleration |
> | -------------------- | ---------- | ------ | ---------- | ----------- | ------------ |
> | ToMA                 | 29.271     | 0.6984 | 0.4258     | 0.7812      | 1.21×        |
> | 8bit                 | 32.331     | 0.8334 | 0.1770     | **1.1971**  | 2.10×        |
> | **8bit‑Z‑Cache (2)** | **33.600** | 0.8289 | **0.1337** | 0.8877      | 3.12×    |
> | 8bit‑TaylorSeer (2)  | 33.085     | 0.8197 | 0.1413     | 0.8773      | 3.12×        |
> | Flux‑mini            | 28.101     | 0.5186 | 0.6449     | 0.2522      | 3.80×        |
> | Z‑Cache (4)          | 30.570     | 0.7758 | 0.2406     | 0.9890  | 2.17×        |
>
> Key observations:
>
> * **Complementarity**: Z‑Cache is orthogonal to quantization. Combining Z‑Cache with 8‑bit (8bit‑Z‑Cache) pushes the speedup to **3.1×** while improving PSNR/LPIPS over 8bit alone.
> * **Gains over the same base**: At the *same* 3.1× acceleration, 8bit‑Z‑Cache clearly outperforms 8bit‑TaylorSeer on **all four quality metrics** (PSNR, SSIM, LPIPS, ImageReward), showing that the reflection step brings non‑trivial benefits even on top of a compressed backbone.
> * **Versus distilled models**: Flux‑mini attains a similar acceleration (3.8×) but with much worse PSNR/SSIM/LPIPS and ImageReward, indicating that cache‑based methods remain competitive with distilled models in the high‑speed regime.
>
> These results support our claim that Z‑Cache is not an alternative to quantization / pruning, but a **complementary plug‑in** that can be layered on top of them.
>
> We will summarize these findings in the rebuttal and move the full table to the appendix in a camera‑ready version if allowed.
>
> *4b. Extension to U‑Net‑based frameworks (SDXL)*
>
> To address the concern that most experiments are on DiT‑style models, we additionally tested Z‑Cache on **SDXL**, a U‑Net‑based diffusion model, and compared it directly to TaylorSeer at the same step budgets:
>
> | Method          | PSNR       | SSIM       | LPIPS      | ImageReward | Acceleration |
> | --------------- | ---------- | ---------- | ---------- | ----------- | ------------ |
> | **Z‑Cache (5)** | **29.399** | **0.7112** | **0.3100** | **0.3788**  | 2.61×        |
> | TaylorSeer (5)  | 29.028     | 0.6969     | 0.3294     | 0.3350      | 2.61×        |
> | **Z‑Cache (7)** | **28.589** | **0.6373** | **0.4162** | **0.1802**  | 4.00×        |
> | TaylorSeer (7)  | 28.251     | 0.6041     | 0.4528     | 0.1024      | 4.00×        |
>
> Across both acceleration settings, Z‑Cache improves PSNR/SSIM, reduces LPIPS, and yields higher ImageReward than TaylorSeer under the **same number of cache steps**, demonstrating that the cache‑then‑check mechanism is effective beyond DiT and naturally carries over to U‑Net architectures.
>
> **5. Hyperparameters: choice of base cache and averaging weight**
>
> We see Z‑Cache as a **framework** on top of any single‑trajectory cache baseline:
>
> * The choice of base cache is analogous to choosing a backbone; Z‑Cache improves each base at a fixed acceleration ratio (Tab. 4 in the paper), and can wrap *future* caching methods without modification.
> * The averaging weight function does not introduce tunable hyperparameters. In all experiments we use the **logarithmic schedule** described in Sec. 3.3, which depends only on the normalized timestep within a window and has **no learned or hand‑tuned scalars**.
>
> In **Figure 9**, we compare logarithmic, linear and sinusoidal choices; the logarithmic schedule yields small but consistent improvements on PSNR/SSIM/LPIPS, which is why we adopt it throughout.   We will make this “hyperparameter‑free” aspect more explicit in the text.
>
> **6. Fig. 5 cases and failure analysis**
>
> Our intention with **Fig. 5** is to illustrate **semantic consistency** under ≈5× acceleration rather than to argue that every pixel is strictly better. In the examples the reviewer mentioned (rows 1–5), we find that:
>
> * Z‑Cache best preserves scene layout and textual elements (e.g., tree placements in first row, shape of rivers in the third row),
> * whereas other methods more frequently drop objects, misplace text, or introduce artifacts, consistent with our superior PSNR/SSIM/LPIPS at the same acceleration.
>
> That said, we agree that **explicit failure analysis** is valuable. In our internal inspection, Z‑Cache behaves similarly to other cache methods when the cache interval is pushed far beyond the values reported in Tables 1–3: small text and very high‑frequency textures can degrade, and the benefit of reflection diminishes.

---

> > ### Comment · Reviewer_bgiZ · 2025-11-25
> >
> > Thanks for the rebuttal. My concerns are addressed.

---

### Official Review · Reviewer_F2wx · 2025-11-01

**Soundness:** 2
**Presentation:** 3
**Contribution:** 2
**Rating:** 4
**Confidence:** 4

**Summary:**

Z-Cache proposes a “cache-then-check” framework to accelerate diffusion Transformers while preserving generation quality. Starting from any existing feature-caching baseline, the method first caches or extrapolates hidden states for the next N timesteps, performs one full forward pass at timestep t+N+1, then rolls back to revisit the N cached steps. The newly computed “future” features are blended with the original predictions through a distance-dependent logarithmic weight, yielding corrected hidden states before sampling continues. Experiments on DiT-XL/2, FLUX-dev and HunyuanVideo show 4.5–6.2× speed-ups.

**Strengths:**

1.	Achieves 4–6× inference acceleration across image and video generation tasks.

2.	Method is lightweight and model-agnostic: it requires no retraining of the backbone.

**Weaknesses:**

1.	The novelty is Limited novelty. The contribution is an incremental refinement over existing cache-and-forecast strategies rather than a fundamentally new acceleration paradigm.

2.	Noticeable quality drops in challenging regions, e.g., facial consistency in Figure 1.

3.	Figures in the paper are not sufficiently high-resolution, making it hard to inspect the visual differences.

**Questions:**

N/A

---

> ### Author Response · Authors · 2025-11-14
>
> We thank the reviewer for the detailed feedback and constructive suggestions.
>
> **1. Novelty and relation to prior cache-and-forecast methods**
>
> Our goal is not only to forecast features more accurately but also to change the structure of the sampling trajectory. Prior cache-and-forecast works (e.g., TaylorSeer, TeaCache) operate on a single forward trajectory: hidden states for intermediate steps are predicted from past states and then directly used for sampling. In contrast, Z-Cache introduces a dual-trajectory, cache-then-check paradigm:
>
> - We first follow a standard cache-then-forecast trajectory.
>
> - After the next full computation, we roll back and re-sample the cached steps by fusing features from both full steps (past and future) with a distance-dependent weight.
>
> This “zig-zag” pattern is qualitatively different from extrapolating along a single trajectory and is what gives rise to the self-reflection behavior highlighted in Fig. 2 and Fig. 4.
>
> Empirically, this design is orthogonal to the base caching rule: across DiT-XL/2, FLUX-dev, and HunyuanVideo, Z-Cache consistently improves the underlying cache baseline (TaylorSeer or TeaCache) at the same acceleration ratio (Tables 1–4), which we believe goes beyond a small tweak of existing methods.
>
> **2. Quality drops in challenging local regions**
>
> We agree that small local degradations (e.g., facial details) are important at high accelerations. Two clarifications:
>
> - **Compression artifacts**. The main paper uses heavily compressed figures due to the submission size constraint, which exaggerates differences. We will upload a high-resolution version.
>
> - **Patch-level analysis already included**. In Fig. 7 we show patch-level comparisons (hair texture, art style). Z-Cache better preserves fine textures relative to other cache baselines under the same speed-up, which is also consistent with our LPIPS improvements in Tables 1 and 2.
>
> **3. Figure resolution**
>
> We will update a high-resolution PDF.

---

### Meta-Review · Area_Chair_tneS · 2026-01-09

**Summary:**

This paper addresses the latency problem with a training-free method that is easy to plug into existing DiT/FLUX/HunyuanVideo models and delivers approximately 4–6× speedups while retaining generative quality. This is a significant issue in diffusion transformer models. Despite the importance of the task, three reviewers gave negative scores and only one reviewer was satisfied with the results.

**Reviewer Concerns:**

Most reviewers raised concerns about the novelty, methodology, semantic errors, experiments, and evaluations. During the rebuttal period, the authors attempted to address these issues, but their responses did not appear to satisfy the reviewers.

The AC also agrees with the reviewers’ concerns. Therefore, I recommend rejecting this paper.

**Reviewer Scores:**

After addressing most concerns in the rebuttal, one score is expected to improve to positive (Reviewer bgiZ: 4 $\rightarrow$ 6). This would result in a total of two positive scores (Reviewers bgiZ and 3euh at 6), while Reviewers F2wx and aZpU would remain at 4.

---

### Decision · Program_Chairs · 2026-01-26

Reject